 eLife

# Genetic variation of putative myokine signaling is dominated by biological sex and sex hormones

**Leandro M Velez**[1,2†], **Cassandra Van**[1,2†], **Timothy Moore**[3], **Zhenqi Zhou**[4], **Casey Johnson**[1,2], **Andrea L Hevener**[4,5*], **Marcus M Seldin**[1,2*]

[1]Department of Biological Chemistry, University of California, Irvine, Irvine, United States; [2]Center for Epigenetics and Metabolism, University of California Irvine, Irvine, United States; [3]Department of Medicine, Division of Cardiology, David Geffen School of Medicine at UCLA, Los Angeles, United States; [4]Department of Medicine, Division of Endocrinology, Diabetes and Hypertension, David Geffen School of Medicine at UCLA, Los Angeles, United States; [5]Iris Cantor-UCLA Women's Health Research Center, David Geffen School of Medicine at UCLA, Los Angeles, United States

**\*For correspondence:**
ahevener@mednet.ucla.edu
(ALH);
mseldin@uci.edu (MMS)

[†]These authors contributed
equally to this work

**Competing interest:** The authors
declare that no competing
interests exist.

**Reviewing Editor:** Christopher
L-H Huang, University of
Cambridge, United Kingdom

**Abstract** Skeletal muscle plays an integral role in coordinating physiological homeostasis, where signaling to other tissues via myokines allows for coordination of complex processes. Here, we aimed to leverage natural genetic correlation structure of gene expression both within and across tissues to understand how muscle interacts with metabolic tissues. Specifically, we performed a survey of genetic correlations focused on myokine gene regulation, muscle cell composition, cross-tissue signaling, and interactions with genetic sex in humans. While expression levels of a majority of myokines and cell proportions within skeletal muscle showed little relative differences between males and females, nearly all significant cross-tissue enrichments operated in a sex-specific or hormone-dependent fashion; in particular, with estradiol. These sex- and hormone-specific effects were consistent across key metabolic tissues: liver, pancreas, hypothalamus, intestine, heart, visceral, and subcutaneous adipose tissue. To characterize the role of estradiol receptor signaling on myokine expression, we generated male and female mice which lack estrogen receptor α specifically in skeletal muscle (MERKO) and integrated with human data. These analyses highlighted potential mechanisms of sex-dependent myokine signaling conserved between species, such as myostatin enriched for divergent substrate utilization pathways between sexes. Several other putative sex-dependent mechanisms of myokine signaling were uncovered, such as muscle-derived tumor necrosis factor alpha (*TNFA*) enriched for stronger inflammatory signaling in females compared to males and *GPX3* as a male-specific link between glycolytic fiber abundance and hepatic inflammation. Collectively, we provide a population genetics framework for inferring muscle signaling to metabolic tissues in humans. We further highlight sex and estradiol receptor signaling as critical variables when assaying myokine functions and how changes in cell composition are predicted to impact other metabolic organs.

## Editor's evaluation

This elegantly performed systems-genetics paper on the predicted human skeletal muscle secretome highlights the importance of sex and sex hormones in regulating myokine expression and predicted cross-tissue effects. Male and female mice lacking estrogen receptor α (Esr1) were used to understand how estrogen signalling affects myokine genes expression. The methods used and data presented in this manuscript can serve as an important resource for other researchers in the field.

**eLife digest** The muscles that are responsible for voluntary movements such as exercise are called skeletal muscles. These muscles secrete proteins called myokines, which play roles in a variety of processes by interacting with other tissues. Essentially, myokines allow skeletal muscles to communicate with organs such as the kidneys, the liver or the brain, which is essential for the body to keep its metabolic balance. Some of the process myokines are involved include inflammation, cancer, the changes brought about by exercise, and even cognition. Despite the clear relevance of myokines to so many physiological outcomes, the way these proteins are regulated and their effects are not well understood.

Genetic sex – specified by sex chromosomes in mammals – contributes to critical aspects of physiology. Specifically, many of the metabolic traits impacted by myokines show striking differences arising from hormonal or genetic interactions depending on the genetic sex of the subject being studied. It is therefore important to consider genetic sex when studying the effects of myokines on the body.

Velez, Van et al. wanted to gain a better understanding of how skeletal muscles interact with metabolic tissues such as pancreas, liver and brain, taking genetic sex into consideration. To do this they surveyed human datasets for the correlations between the activity of genes that code for myokines, the composition of muscle cells, the signaling between muscles and metabolic tissues and genetic sex.

Their results showed that, genetic sex and sex hormones predicted most of the effects of skeletal muscle on other tissues. For example, myokines from muscle were predicted to be more impactful on liver or pancreas, depending on whether individuals were male or female, respectively.

The results of Velez, Van et al. illustrate the importance of considering the effects of genetic sex and sexual hormones when studying metabolism. In the future, these results will allow other researchers to design sex-specific experiments to be able to gather more accurate information about the mechanisms of myokine signaling.

## Introduction

Proteins secreted from skeletal muscle, termed myokines, allow muscle to impact systemic physiology and disease. Myokines play critical roles in a variety of processes, including metabolic homeostasis, exercise improvements, inflammation, cancer, and cognitive functions (*Severinsen and Pedersen, 2020*; *Eckel, 2019*; *Febbraio and Pedersen, 2020*; *Kim et al., 2019*; *Kim et al., 2021*; *Seldin and Wong, 2012*). Several notable examples include key peptide hormones such as myostatin and interleukin-6 which exert potent actions in regulating autocrine/paracrine muscle physiology (*Kollias and McDermott, 2008*) and beneficial exercise-induced endocrine signaling (*Severinsen and Pedersen, 2020*), respectively. Despite the clear relevance of these factors in mediating a multitude of physiological outcomes, the genetic architecture, regulation, and functions of myokines remain inadequately understood. Given that genetic sex contributes critically to nearly every physiological outcome, it is essential to consider when relating specific mechanisms to complex genetic and metabolic interactions. Specifically, many metabolic traits impacted by myokines show striking sex differences arising from hormonal (*Mauvais-Jarvis et al., 2017*; *Mauvais-Jarvis, 2015*; *Clegg and Mauvais-Jarvis, 2018*; *Ribas et al., 2016*), genetic (*Mauvais-Jarvis et al., 2017*; *Zore et al., 2018*), or gene-by-sex interactions (*Norheim et al., 2019*; *Chella Krishnan et al., 2021*). In this study, we leveraged natural genetic correlation structure of gene expression both within and across tissues to understand how muscle interacts with metabolic tissues. Collectively, we provide a population genetics framework for inferring muscle signaling to metabolic tissues in humans. We further highlight sex and estradiol receptor signaling as critical variables when assaying myokine functions and how changes in cell composition are predicted to impact other metabolic organs.

## Results

### Sex hormone receptors are enriched with myokine expression independent of biological sex

Our goal was to exploit correlation structure of natural genetic variation to investigate how skeletal muscle communicates with and impacts metabolic organs. We first assayed regulation of myokines and changes in cellular composition, then related these observations to inferred cross-tissue signaling mechanisms (*Figure 1A*). Initially, we performed differential expression of genes encoding all known secreted proteins (3666 total) in skeletal muscle from 210 male and 100 female individuals (*Battle et al., 2017*). While several notable myokines appeared different between sexes (*Figure 1B*), a striking majority of all secreted proteins (74%) showed no difference in expression between males and females (*Figure 1C*, *Supplementary file 1*). To understand potential sex effects on the regulation of myokines, gene ontology (GO) enrichments were performed on genes which showed the strongest correlation with myokines corresponding to each category (male-specific, female-specific, or non-sex-specific). Here, the top 10 pathways which persisted in females were also always observed to overlap with the non-sex-specific category (*Figure 1D*). In contrast, pathways enriched for male-specific myokines were distinct (*Figure 1D*). Notably, the female and shared pathways suggested roles in epigenetics and RNA processing, while male-specific myokine coregulated processes were more enriched in metabolic pathways (e.g. NADH metabolism) (*Figure 1D*). Further, a majority of myokines showed strong correlation with receptors mediating functions of androgens (androgen receptor [AR]), estradiol (estrogen receptor α – ESR1), or both, regardless of sex-specific expression (*Figure 1E*). We note that expression of hormone receptors themselves were also not significantly different between sexes (*Figure 1—figure supplement 1*). To infer causality from hormone receptor regulation, we performed RNA-sequencing (RNA-Seq) on mice lacking *Esr1* in skeletal muscle specifically (MERKO) and integrated these analyses with human myokine estimates. While myokines not regulated by *Esr1* showed little sex-specific differences in expression, those which were estrogen-dependent showed much stronger representation of sex specificity, in particular in males (*Figure 1F–G*). Among these was the master regulator of skeletal muscle differentiation and proliferation, myostatin (MSTN), where hormone receptor correlations and gene expression were markedly higher in males compared to females (*Figure 1H*). Further, ablation of *Esr1* in mice uniquely drove expression changes in males (*Figure 1H*). These data suggest interactions between biological sex and ESR1 to tightly regulate *MSTN* in males, where other factors could contribute more in females. Given that, like many bioactive secreted proteins, the regulation and sex specificity of myostatin are additionally controlled via post-transcriptional mechanisms (*McMahon et al., 2003*), we next explored gene expression changes at the protein level. Immunoblots were performed on skeletal muscle from male and female WT or MERKO mice (*Figure 1—figure supplement 2*). Quantification of the processed form of myostatin showed that, consistent with the RNA-Seq in mice and humans, the protein trended toward higher levels in male compared to female mice, where ablation of *Esr1* showed a reduction (*Figure 1I*). Dissimilar to the mouse sequencing data but consistent with human correlations, female MERKO mice showed a reduction in processed form of myostatin relative to their WT controls (*Figure 1I*). Related to the sex-specific regulation of myostatin observed at both RNA and protein levels, gene expression also showed differences in functional annotations. Here, the most highly enriched pathways in males showed GO terms related to glycolytic metabolism (*Figure 1J*) compared to oxidative phosphorylation in females (*Figure 1K*). These observations are consistent with previous studies which note myostatin-dependent increases in muscle mass in males, but not females (*McMahon et al., 2003*; *Reisz-Porszasz et al., 2003*), where estradiol signaling is suggested as a mechanism mediating these differences. These data demonstrate that, expression of most myokines are not different between genetic sexes; however, interactions between sex and hormone receptors likely play important roles in determining myokine regulation and local signaling.

### Sex dominates cross-tissue pathways enriched for myokines

Given that expression levels of most myokines appeared similar between sexes, we next assessed putative functions across organs. We applied a statistical method developed to infer cross-tissue signaling which occur as a result of genetic variation (*Seldin et al., 2018*; *Seldin and Lusis, 2019*; *Seldin et al., 2019*). Here, we assayed the distribution of midweight bicorrelation coefficients between myokine expression levels and global gene expression in key metabolic tissues including hypothalamus, heart,

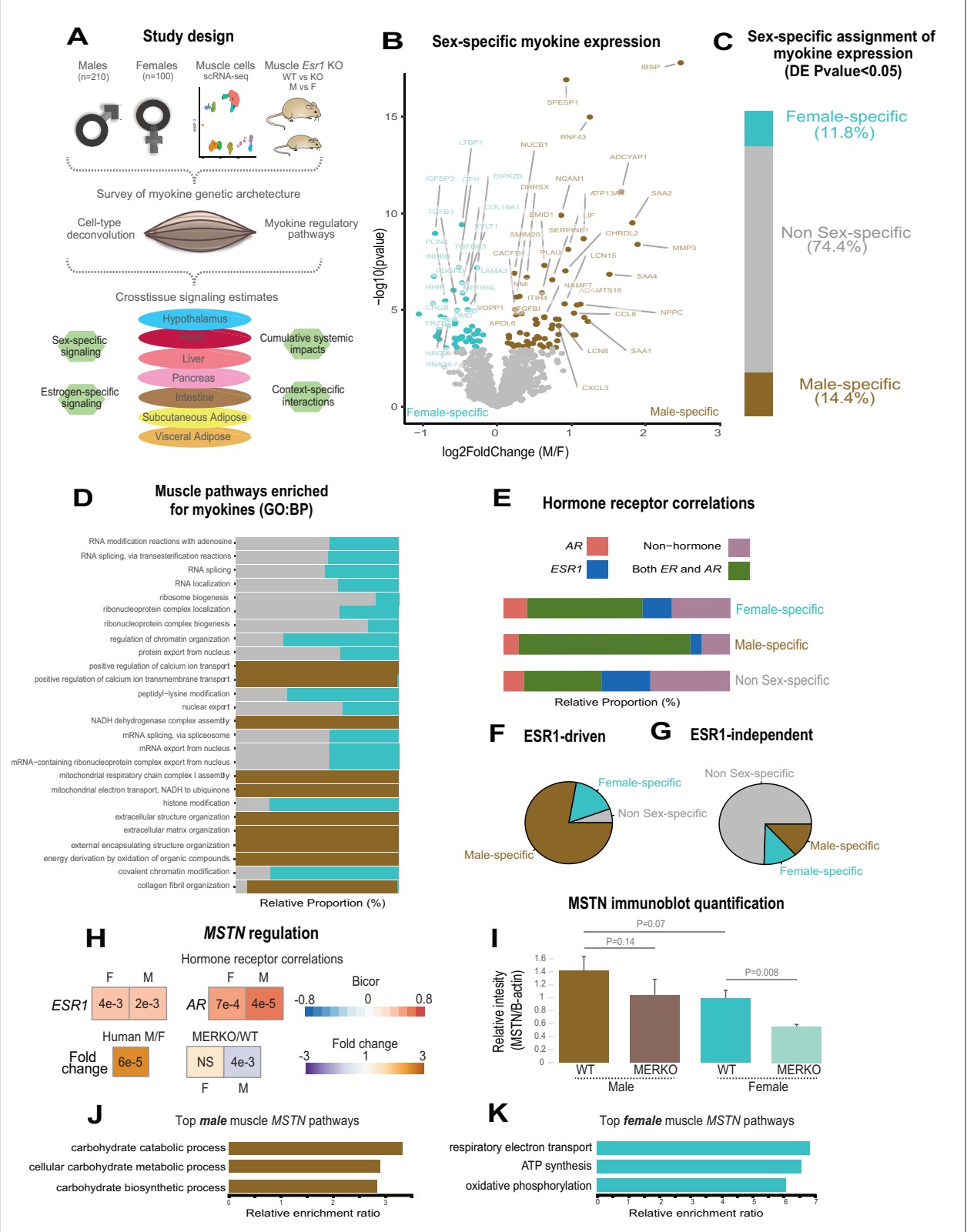

**Figure 1.** Sex and hormone effects on myokine regulation. (**A**) Overall study design for integration of gene expression from muscle from 310 humans, single-cell RNA-sequencing (RNA-seq), muscle-specific deletion of *Esr1* to infer interorgan coregulatory process across major metabolic tissues. (**B–C**) Differential expression analysis for sex was performed on all genes corresponding to secreted proteins in skeletal muscle (myokines). The specific genes which showed significant changes in each sex are shown as a volcano plot (**B**) and the relative proportions of myokines corresponding to each category

*Figure 1 continued on next page*

*Figure 1 continued*

at a least-stringent logistic regression p-value less than 0.05 (**C**). (**D**) For each differential expression category based on sex shown in C, myokines were correlated with all other muscle genes for pathway enrichment. Then the top 10 enriched pathways in males, females, or non-sex specific (by overall significance) were visualized together where number of genes corresponding to each category shown as a relative proportion. (**E**) The same analysis as in D, except instead of myokines being correlated with *AR, ESR1*, both hormone receptors, or neither, as compared to correlating with all genes. (**F–G**) Myokines were binned into two categories based on significant differential expression (logistic regression adjusted p-value < 0.05) between muscle-specific WT and MERKO mice (**F**) or those that showed no change (**G**), then visualized as relative proportions within each category shown in (**C**). (**H**) Midweight bicorrelation (bicor) coefficients (color scheme) and corresponding regression p-values (filled text) are shown for muscle *MSTN ~ ESR1* or *AR* in both sexes (top). Below, correlations are shown for differential expression log2FC (color scheme) and corresponding logistic regression p-values (text fill) for *MSTN* between sexes in humans or WT vs. MERKO mice. (**I**) Quantification of processed form of myostatin (*Figure 1—figure supplement 2*, bottom band) relative to β-actin in WT or MERKO muscle in male or female mice. p-Values calculated using a Student's t-test. (**J–K**) The top three pathways of genes which significantly (p < 1e-4) correlated with muscle *MSTN* in males (**J**) or females (**K**). For human data, n = 210 males and n = 100 females. For mouse MERKO vs. WT comparisons, n = 3 mice per group per sex. p-Values from midweight bicorrelations were calculated using the Student's p-value from WGCNA and logistic regression p-values were calculated using DESeq2.

The online version of this article includes the following source data and figure supplement(s) for figure 1:

**Source data 1.** Skeletal muscle sex hormone receptor expression between sexes.

**Figure supplement 1.** Skeletal muscle sex hormone receptor expression between sexes.

**Figure supplement 2.** Immunoblot for myostatin in EDL muscle from WT and MERKO male and female mice.

intestine, pancreas, liver, subcutaneous, and visceral adipose tissue. Remarkably, nearly all highly significant correlations between myokines and target organ genes (putative direct interactions) showed sex-specific modes of operation (*Figure 2A–H*). This sex specificity also appeared more pronounced for positive correlations between myokines and target tissue genes, as compared to negative (*Figure 2H*). Further, among these significant cross-tissue circuits, hormone receptor enrichments for these myokines were strongly dependent on the category (e.g. significant only in females) rather than target tissue (*Figure 2A–H*). This observation further suggests that hormone receptor levels (ESR1 or AR) in muscle are a stronger determinant of myokine expression compared to genetic sex; however, sex is suggested to dominate coregulated signaling processes across organs via myokines. To gauge the relative impact of muscle steroid hormone receptors across organs, the number of significant correlations between *ESR1, AR,* or both were quantified from muscle to each tissue. Here, *ESR1* showed an order of magnitude stronger enrichment across metabolic tissues compared to *AR* or correlation with both hormone receptors (*Figure 2I–J*). Additionally, the number of significantly correlated cross-tissue male *ESR1* genes (*Figure 2I*) was threefold higher than females (*Figure 2J*). Because both sex and ESR1 signaling appeared to contribute to the regulation and functions of myokines, significant cross-tissue enrichments were binned into categories taking into consideration whether myokines were driven by ESR1 in muscle, and/or showing a sex-specific mode of cross-organ significance. This analysis suggested that a majority of myokines were either driven by ESR1 and signaled robustly across sexes (*Figure 2K*, yellow) or signaled differently between sexes, but regulated independent of ESR1 (*Figure 2K*, red). These categories appeared to a much greater extent compared to a combination of both *ESR1*-driven myokine and sex-specific cross-tissue signaling (*Figure 2K*, beige) or neither (*Figure 2K*, seagreen). One notable example of predicted sex-specific signaling was observed for tumor necrosis factor alpha (TNFA). When compared between sexes, muscle *TNFA* showed markedly different putative target tissues (*Figure 2L*, left), as well as underlying functional pathways (*Figure 2L*, right). For example, overall inflammatory processes engaged by TNFA were stronger in adipose tissue in females; however, the same pathways were higher in liver and hypothalamus in males (*Figure 2L*, left). Collectively, these data show that genetic sex and related sex steroid hormones, particularly estradiol, exert dominant roles in regulating predicted tissue and pathway engagement by myokines.

## Muscle cell proportions are similar between sexes, but associated changes across tissues show sex specificity

To determine the potential impact of muscle composition on other tissues, we next surveyed muscle cellular proportions in the context of genetics and sex. Single-cell sequencing of human skeletal muscle (*Rubenstein et al., 2020*) was integrated using cellular deconvolution (*Danziger et al., 2019*) to roughly estimate cellular composition in the population (*Figure 3A*). Here, a proportion in admixture approach (*Langfelder and Horvath, 2008*) outperformed other methods (*Figure 3—figure*

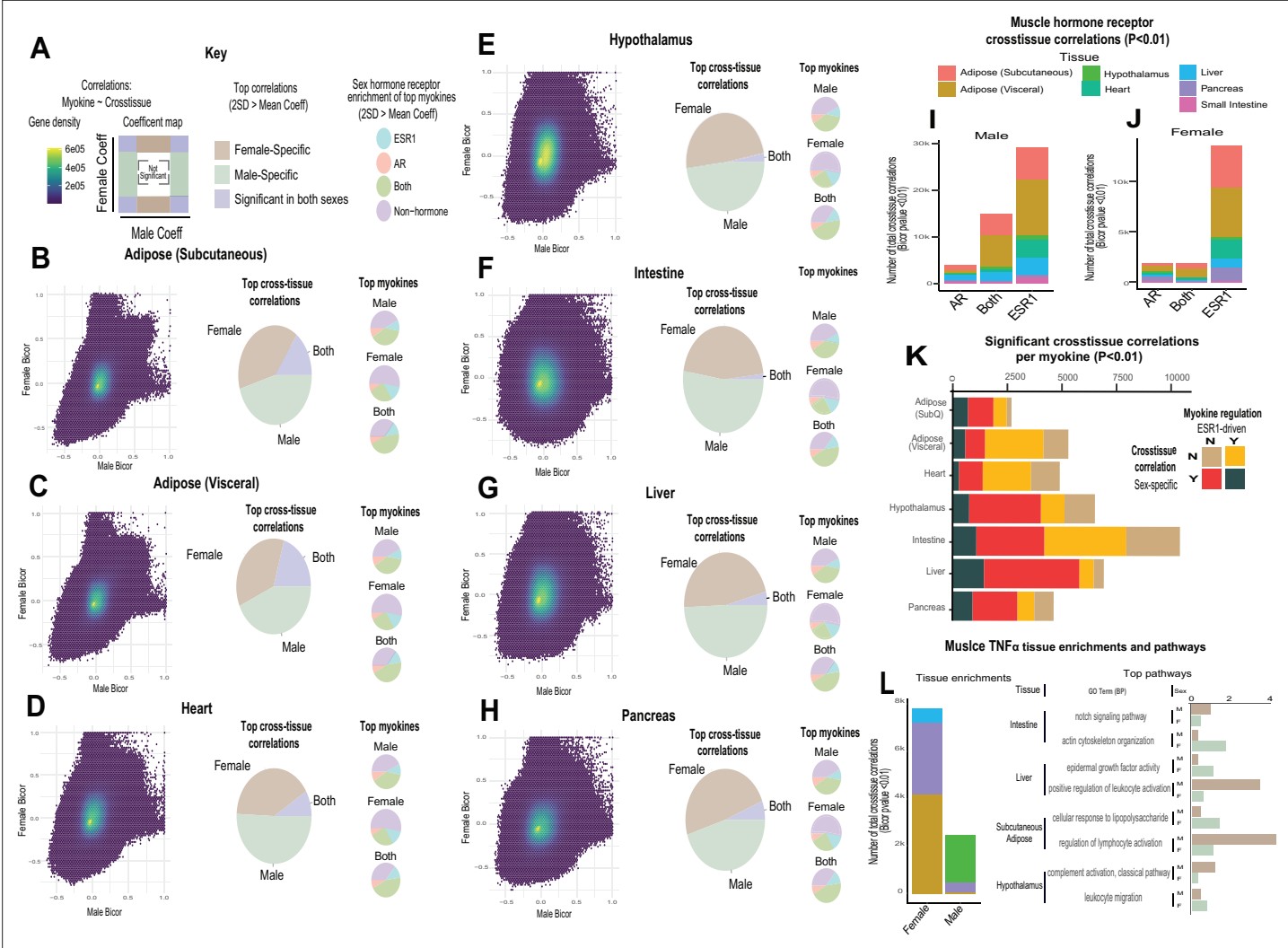

**Figure 2.** Sex and hormone effects on myokine regulation. (A–H) Key illustrating analysis for distribution of midweight bicorrelation coefficients between all myokines in skeletal muscle and global transcriptome measures in each target tissue. Coefficients are plotted between sexes (left), where proportions for 2SD > mean are subdivided into occurrence uniquely in females, males, or shared (middle). The significant (2SD > mean) myokines identified in each category were then binned into hormone receptor correlations for *ESR1*, *AR*, both, or neither (right). This analysis was performed on all myokines across subcutaneous adipose tissue (B), visceral adipose (C), heart (D), hypothalamus (E), small intestine (F), liver (G), and pancreas (H). (I–J) Significant cross-tissue correlations between muscle *ESR1*, *AR*, or both hormone receptors are colored by tissue and shown for males (I) or females (J). (K) For each tissue (y-axis), the ratio of significant cross-tissue correlations per muscle myokine (x-axis) are shown and colored by categories of either the myokine regulated by *ESR1* and/or a significant target tissue regression occurring specifically in one sex. (L) Number of significant cross-tissue correlations with muscle *TNFα* are shown for each sex and colored by tissue as in I–L (left). The −log10(p-value) of significance in an overrepresentation test (x-axis) are shown for top significant inter-tissue pathways for muscle TNFα in each sex (right).

supplement 1) to capture a majority of established cell populations across individuals (*Supplementary file 2*). Similar to myokine expression, no notable differences were observed between sexes in terms of cell composition, with the exception of modest higher glycolytic fiber in males, compared to elevated oxidative fiber levels in females (*Figure 3B*). Additionally, no differences were observed in the correlations within muscle between compositions (*Figure 3—figure supplement 2*); however, nearly every cross-tissue enrichment corresponding to an individual muscle cell type differed between sexes (*Figure 3C*). Generally, differences in skeletal muscle cell abundance was associated with changes in liver and visceral adipose tissue pathways in males, compared to pancreas in females (*Figure 3C*). In contrast to general myokine enrichments, cell proportions showed stronger correlations with *AR* when compared to *ESR1* across both sexes where the most abundant cell types were significantly enriched for both steroid hormone receptors (*Figure 3D*). To uncover potential direct mechanisms

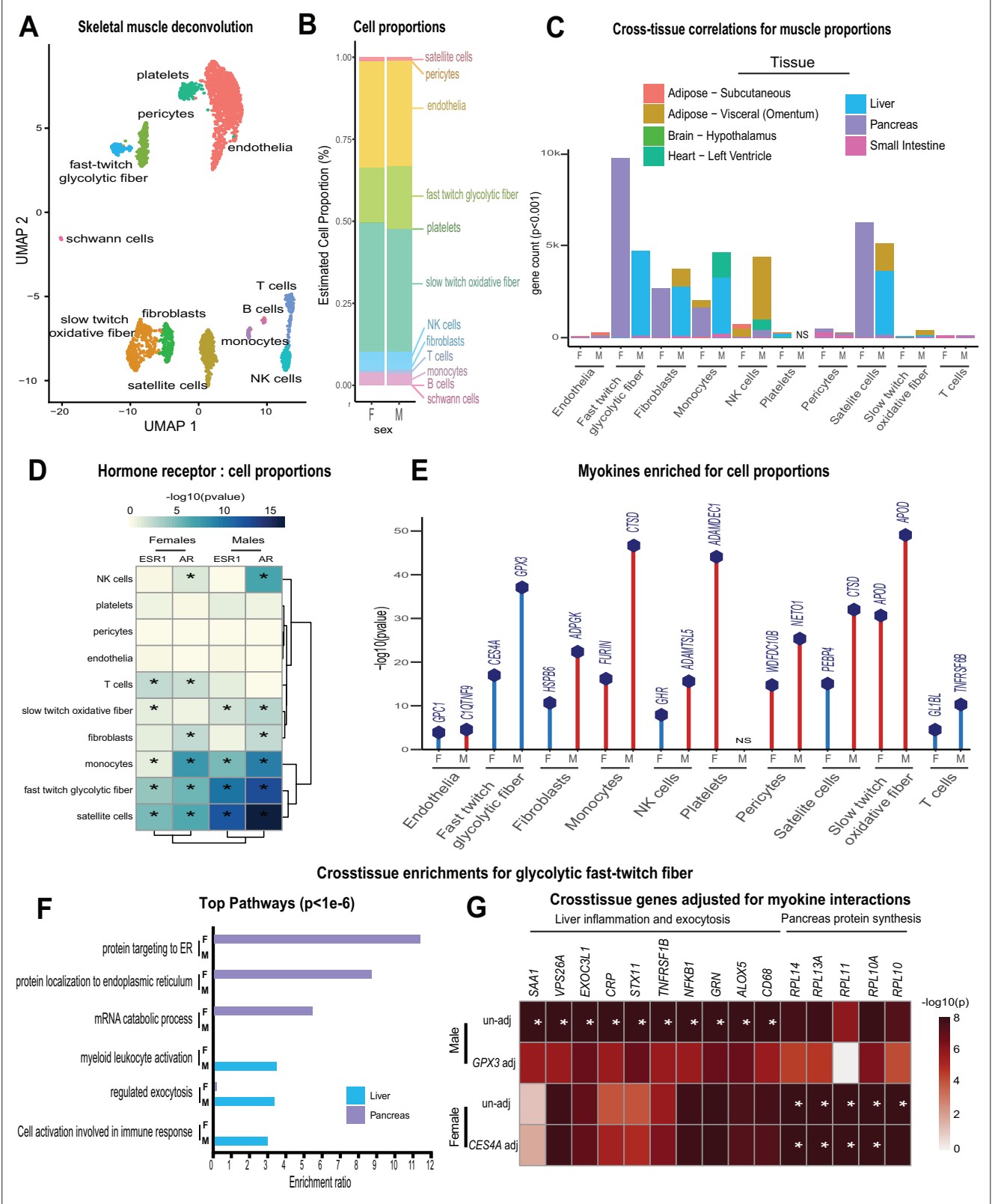

**Figure 3.** Genetic variation of muscle cell proportions and coregulated cross-tissue processes. (**A**) Uniform Manifold Approximation and Projection (UMAP) for skeletal muscle single-cell sequencing used to deconvolute proportions. (**B**) Mean relative proportions of pseudo-single-cell muscle cell compositions (denoted by color) between sexes. (**C**) Number of significant cross-tissue correlations (y-axis) corresponding to each skeletal muscle type in each sex (x-axis). Target tissues are distinguished by color, where NS (male platelets) denotes that no significant cross-tissue correlations were

*Figure 3 continued on next page*

*Figure 3 continued*

observed. (**D**) Heatmap showing significance of correlations between skeletal muscle hormone receptors and cell proportions, * = p < 0.01. (**E**) The strongest enriched myokines are plotted for each myokine (y-axis, −log10 p-value of myokine ~ cell composition) are shown for each muscle proportion for each sex (x-axis). Gene symbols for myokines are shown above each line, where red lines indicate positive correlations between myokine and cell type and blue shows inverse relationships. (**F**) Significant cross-tissue correlated genes in liver (blue) and pancreas (purple) for muscle fast-twitch glycolytic fibers (p < 1e-6) were used for overrepresentation tests where enrichment ratio of significance (x-axis) is shown for each pathway and sex (y-axis). (**G**) Heatmap showing the regression significance of the top five genes corresponding to inflammation (liver), exocytosis (liver), and protein synthesis (pancreas) for proportions of fast-twitch fiber type (un-adj). Below each correlation between fast-twitch fiber and liver or pancreas gene, the same regressions were performed while adjusting for abundance of select myokines in each sex. * = p < 1e-6.

The online version of this article includes the following figure supplement(s) for figure 3:

**Figure supplement 1.** Comparisons of deconvolution methods.

**Figure supplement 2.** Cell composition correlations within each sex.

linking changes in cell composition to peripheral tissues, we correlated all myokines with cell composition profiles. Again, despite few differences between sexes in terms of myokine expression and cell composition, specific myokines highly correlating with individual cell types were markedly different between males and females with the exception of one, *APOD* in slow-twitch fibers (*Figure 3E*). To determine if variation in cell compositions corresponding to sex-specific tissue signaling via myokines was predicted to be likely, we implemented adjusted regression mediation analyses (*Yokota et al., 2020*) for glycolytic fiber composition. Because male glycolytic fiber-type abundance was selectively enriched for liver pathways such as immune cell activation and regulated exocytosis (*Figure 3F*), the top genes driving these enrichments were used for mediation. The top-correlated muscle secreted protein with male glycolytic fiber-type levels was secreted glutathione peroxidase 3 (GPX3). Here, adjusting regressions between glycolytic fiber and liver pathways on *GPX3* reduced the overall significance across tissues (*Figure 3G*), suggesting GPX3 as a potential mediator of this communication. These data point to a potential mechanism whereby muscle fiber abundance could buffer free radical generation in the liver, thereby feeding back on inflammation. This analysis appeared additionally sensitive to inferring non-dependent relationships between muscle cell types, top-ranked myokines, and cross-tissue processes. For example, female glycolytic fibers were strongly enriched for pancreatic protein synthesis pathways; however, when adjusted for the top-ranked myokine *CES4A*, no changes in regression significance were observed (*Figure 3F–G*). These analyses suggest that male GPX3 is a potential mechanism whereby fast-twitch muscle signals to liver; however, the same cell type in females are predicted to drive pancreas protein synthesis independent of CES4A. In summary, we show that cell composition is strongly conserved between sexes, but putative cross-tissue signaling of altered composition differs entirely. We further suggest putative myokines and mechanisms, as well as highlight the key regulatory roles of estrodiol in both sexes.

## Discussion

Here, we provide a population survey of skeletal muscle myokine regulation and putative functions using genetic variation and multi-tissue gene expression data. We find that in general, expression of myokines do not significantly differ between sexes; however, inferred signaling mechanisms across tissues using regressions show strong sex specificity. Steroid hormone receptors, in particular *ESR1*, is highlighted as a key regulator of myokines and potentially interacting with biological sex for proteins such as myostatin. Further integration with loss-of-function mouse models of *Esr1* highlighted the key roles of estradiol signaling in muscle in terms of myokine regulation and signaling across both sexes. Generation of pseudo-single-cell maps of muscle composition showed that, like myokines, muscle proportions are conserved between sexes, but inferred interorgan consequences differ substantially. When interpreting these findings, several considerations should be taken. While inter-tissue regression analyses have been informative to dissect mechanisms of endocrinology (*Seldin et al., 2018*; *Seldin and Lusis, 2019*; *Seldin et al., 2019*; *Li and Auwerx, 2020*), observations can be subjected to spurious or latent relationships in the data. While causality for inter-organ signaling can be inferred statistically using approaches such as mediation as in *Figure 3H*, the only methods to provide definitive validation for new mechanisms are in experimental settings. Further, our current analyses rely on gene expression to guide functions of proteins which are typically strongly regulated

by post-transcriptional processes. As shown for myostatin (*Figure 1*), gene expression analyses can miss key functional aspects of proteins, where follow-up studies and resources focused on protein and subsequent modification levels could heavily improve predictions. In addition, we anticipate that estimates for ESR1 effects on myokines in this study likely represent an underestimated number of all human ESR1-driven myokines. One limitation here includes that annotation of known orthologous mouse-human genes (*The Alliance of Genome Resources Consortium, 2020*) remains somewhat limited. Furthermore, cell composition estimates from single-cell sequencing data are inferred from gene expression, where histological or flow cytometry-based methods can provide much more accurate direct quantifications. Clearly, morphological and structural differences between sexes have been observed in humans (*Haizlip et al., 2015*) which, if not apparent in deconvoluted gene expression, would be missed in this analysis. Future studies addressing these points will help to clarify context- and mechanism-relevant muscle-derived endocrine communication axes. In summary, this study highlights the key contributions of sex and sex steroid hormones in mediating myokine functions.

# Materials and methods

All datasets used, R scripts implemented for analyses, and detailed walkthrough guide is available via: https://github.com/Leandromvelez/myokine-signaling, (copy archived at swh:1:rev:530dc39df-1c586ad67eab36688fa2c1936b06354; *Velez and Seldin, 2022*).

**Key resources table**

| Reagent type (species) or resource | Designation | Source or reference | Identifiers | Additional information |
| --- | --- | --- | --- | --- |
| Antibody | Anti-MSTN (Goat polyclonal) | R&D | AF788 | (1:1000) |
| Antibody | Rabbit anti-βactin (Rabbit polyclonal) | Genetex | GTX109639 | (1:1000) |

## Animals

All mice used in this study were approved by the University of California Los Angeles (UCLA) Animal Care and Use Committee, in accordance with Public Health Service guidelines with reference #92-169.

## Data sources and availability

All data used in this study can be immediately accessed via GitHub to facilitate analysis. Human skeletal muscle and metabolic tissue data were accessed through GTEx V8 downloads portal on August 18, 2021, and previously described (*Battle et al., 2017*). To enable sufficient integration and cross-tissue analyses, these data were filtered to retain genes which were detected across tissues where individuals were required to show counts >0 in 1.2e6 gene-tissue combinations across all data. Given that our goal was to look across tissues at enrichments, this was done to limit spurious influence of genes only expressed in specific tissues in specific individuals. Post-filtering consists of 310 individuals and 1.8e7 gene-tissue combinations. Single-cell sequencing from skeletal muscle used for deconvolution was obtained from *Rubenstein et al., 2020*. *Esr1* WT and KO mouse differential expression results are available on GitHub as well, where raw sequencing data has been deposited in NIH sequence read archive (SRA) under the project accession: PRJNA785746.

## Selection of secreted proteins

To determine which genes encode proteins known to be secreted as myokines, gene lists were accessed from the Universal Protein Resource which has compiled literature annotations terms for secretion (*The UniProt Consortium, 2021*). Specifically, the query terms to access these lists were: locations:(location:"Secreted [SL-0243]" type:component) AND organism:"*Homo sapiens* (Human) [9,606]" where 3666 total entries were found.

## Differential expression of myokines dependent on sex

Gene expression counts matrices were isolated from the rest of the tissues, where individual genes were retained if the total number of counts exceeded 10 in 50 individuals. Next, only genes encoding secreted proteins (above) were retained, where logistic regression contrasted on sex was performed using DESeq2. Differential expression summary statistics were used for downstream binning of sex

specificity based on an empirical logistic regression p-value < 0.05. This threshold was used to reflect a least stringent cutoff where, despite potential false positive influence, genes which nominally trended toward sex-specific expression could be included in those categories. Given that the general conclusions supported very few proportions of myokines showing sex-specific patterns of expression, this conclusion would only be further exaggerated if the differential expression threshold were made more stringent and lessened the number of myokines in each category.

## Regression analyses across tissues

Regression coefficients and corresponding p-values across tissues were generated using WGCNA bicorandpvalue() function (*Langfelder and Horvath, 2008*). Myokine-target gene pairs were considered significant (e.g. *Figure 2A–H*) at a threshold of abs(bicor) > 2 standard deviations beyond the average coefficient for the given target tissue of interest. In previous studies, this threshold of 2 standard deviations reflects adaptive permutation testing p-values < 0.01 (*Seldin et al., 2018*; *Seldin and Lusis, 2019*). For analyses estimating cumulative patterns of concordance across tissues (e.g. *Figure 2I–L*), empirical regression p-values (Student's p-value from bicor coefficients) of 0.01 (corresponding to abs(bicor) > 0.1) were used to assay global patterns. While usage of empirical p-values are clearly subjected to false positives, these were used for several analyses to capture broad visualization of both potential direct interactions which would show significance across multiple FDR thresholds (e.g. myokine-target gene), as well as coregulated indirect processes across organs. Thus, assessing cumulative changes as a result of larger physiologic shifts. It is important to note that we exclusively rely on these empirical p-values when surveying broad correlation structures, whereas much more stringent and appropriate thresholds (e.g. p < 1e-6 for *Figure 3G*) were applied when inferring direct interactions.

## Pathway enrichment analyses

For *Figures 1I and 3G*, genes corresponding to p-value cutoffs were visualized using Webgestalt (*Liao et al., 2019b*) to enable streamline analysis. This tool enabled simultaneous overrepresentation testing of GO:BP (non-redundant), KEGG, and Panther databases. For *Figure 1D*, the top 1000 (by regression p-value) significant genes from myokines to all muscle bicorrelation analysis in females, males, or non-sex specific datasets were assessed for enrichment in GO Biological Process terms using ClusterProfiler v4.0.2 in R (*Wu et al., 2021*). The resulting top 10 GO terms in each dataset were integrated and plotted against the relative proportion of the p-adjusted value and visualized in the same graph using ggplot2.

## Deconvolution of skeletal muscle

Raw single-cell RNA-sequencing from skeletal muscle was obtained from *Rubenstein et al., 2020*. These raw counts were analyzed in Seurat where cluster analyses identified variable cell compositions. Cell type annotations were assigned based on the top 30 genes (*Supplementary file 2*) assigned to each Uniform Manifold Approximation and Projection (UMAP) cluster through manual inspection and ENRICHR (*Kuleshov et al., 2016*). Finally, a normalized matrix of gene:cells was exported from Seurat and used to run deconvolution on skeletal muscle bulk sequencing. Using the ADAPTS pipeline (*Danziger et al., 2019*), three deconvolution methods (nnls, dcq, or proportions in admixture) were compared based on ability to robustly capture cell proportions (*Figure 3—figure supplement 1*), where proportion in admixture showed the best performance and subsequently applied to bulk sequencing.

## ESR1 muscle KO generation, RNA-Seq, and integration with human data

Muscle-specific Esr1 deletion was generated and characterized as previously described (*Ribas et al., 2016*). Whole quadriceps was pulverized at the temperature of liquid nitrogen. Tissue was homogenized in Trizol (Invitrogen, Carlsbad, CA), RNA was isolated using the RNeasy Isolation Kit (Qiagen, Hilden, Germany), and then tested for concentration and quality with samples where RIN > 7.0 used in downstream applications. Libraries were prepared using KAPA mRNA HyperPrep Kits and KAPA Dual Index Adapters (Roche, Basel, Switzerland) per manufacturer's instructions. A total of 800–1000 ng of RNA was used for library preparation with settings 200–300 bp and 12 PCR cycles. The resultant

libraries were tested for quality. Individual libraries were pooled and sequenced using a HiSeq 3000 at the UCLA Technology Center for Genomics and Bioinformatics (TCGB) following in-house established protocols. Raw RNA-Seq reads were inspected for quality using FastQC v0.11.9 (Barbraham Institute, Barbraham, England). Reads were aligned and counted using the Rsubread v2.0.0 (*Liao et al., 2019a*) package in R v3.6 against the Ensembl mouse transcriptome (v97) to obtain counts. Lowly expressed genes (>80% samples with 0 count for particular gene) were removed. Samples were analyzed for differential expression using DeSeq2 v1.28.0 (*Love et al., 2014*).

## Conservation of gene between mice and humans

To find which myokines and pathways were conserved between mice and humans, all orthologous genes were accessed from MGI vertebrate homology datasets, which have been compiled from the Alliance for Genome Resources (*The Alliance of Genome Resources Consortium, 2020*) and intersected at the gene level (roughly 18,000 genes).

## Immunoblotting procedures

Muscle tissue was homogenized in the TissueLyser II (Qiagen) at 4°C in RIPA lysis buffer supplemented with protease inhibitors. The homogenate was centrifuged at 4°C for 10 min at 10,000 *g*, and the protein concentrations in the supernatant were measured by the BCA assay (Bio-Rad). After boiling protein samples for 5 min, 20 µg of protein from each sample were applied on an SDS-polyacrylamide gel (10%) and electrophoresis was performed at 100 V for 1.5 hr. The separated proteins were transferred to nitrocellulose membranes and membranes were blocked for 1.5 hr in TBS (4 mM Tris-HCl, pH 7.5, and 100 mM NaCl) containing 5% skim milk plus Tween 20, at room temperature. Goat polyclonal anti-GDF8 (Myostatin) (R&D, catalog number AF788) at 1/1000 dilution were applied overnight as primary antibody. After washings, membranes were incubated with Goat IgG HRP-conjugated Antibody (R&D HAF017) at 1/10,000 for 2 hr, and bound HRP activity was detected with an enhanced chemiluminescence method (Clarity Western ECL, Bio-Rad), by means of a chemiluminescence detection system (ChemiDoc System, Bio-Rad). The intensities of the resulting bands were quantified by densitometry (ImageJ free software). Membranes were immersed in a stripping solution for 10 min (Restore PLUS Western Blot, Thermo Fisher), and then the process repeated with a rabbit polyclonal anti-β-actin (GeneTex GTX109639) at 1/1000 dilution as loading control to assess uniformity of loading.

## Acknowledgements

We acknowledge the following funding sources for supporting these studies: LMV, CV, CJ, and MMS were supported by NIH grants HL138193, DK130640, and DK097771. ZZ was supported by NIH grant DK125354. TMM was supported by the UCLA Intercampus Medical Genetics Training Program (T32GM008243). ALH is supported by NIH grants U54 DK120342, R01 DK109724, and P30 DK063491.

## Additional information

### Funding

| Funder | Grant reference number | Author |
| --- | --- | --- |
| National Institutes of Health | R00: HL138193 | Leandro M Velez<br>Cassandra Van<br>Casey Johnson<br>Marcus M Seldin |
| National Institutes of Health | DP1: DK130640 | Leandro M Velez<br>Cassandra Van<br>Casey Johnson<br>Marcus M Seldin |
| National Institutes of Health | dkNET pilot grant: DK097771 | Leandro M Velez<br>Cassandra Van<br>Casey Johnson<br>Marcus M Seldin |

| Funder | Grant reference number | Author |
|---|---|---|
| National Institutes of Health | UCLA Intercampus Medical Genetics Training Program: T32GM008243 | Timothy Moore |
| National Institutes of Health | U54: DK120342 | Andrea L Hevener |
| National Institutes of Health | R01: DK109724 | Andrea L Hevener |
| National Institutes of Health | P30: DK063491 | Andrea L Hevener |
| National Institutes of Health | R01: DK125354 | Zhenqi Zhou |

The funders had no role in study design, data collection and interpretation, or the decision to submit the work for publication.

## Author contributions
Leandro M Velez, Conceptualization, Formal analysis, Investigation, Writing – original draft; Cassandra Van, Data curation, Formal analysis, Methodology, Software, Writing – original draft, Writing – review and editing; Timothy Moore, Data curation, Formal analysis, Investigation, Resources, Writing – original draft, Writing – review and editing; Zhenqi Zhou, Methodology, Resources, Supervision; Casey Johnson, Conceptualization, Methodology, Validation, Writing – review and editing; Andrea L Hevener, Investigation, Methodology, Resources, Writing – original draft; Marcus M Seldin, Conceptualization, Formal analysis, Funding acquisition, Investigation, Methodology, Supervision, Visualization, Writing – original draft

## Author ORCIDs
Leandro M Velez  http://orcid.org/0000-0001-8371-2633
Cassandra Van  http://orcid.org/0000-0002-8434-0189
Andrea L Hevener  http://orcid.org/0000-0003-1508-4377
Marcus M Seldin  http://orcid.org/0000-0001-8026-4759

## Ethics
All animal research was approved by the UCLA IACUC, where dare and procedures described in detail here: 10.1126/scitranslmed.aax8096.

## Decision letter and Author response
Decision letter https://doi.org/10.7554/eLife.76887.sa1
Author response https://doi.org/10.7554/eLife.76887.sa2

# Additional files

## Supplementary files
- Transparent reporting form
- Supplementary file 1. DE statistics for myokines based on sex.
- Supplementary file 2. Cell type marker genes used.

## Data availability
All datasets and detailed analysis available at: https://github.com/Leandromvelez/myokine-signaling, (copy archived at swh:1:rev:530dc39df1c586ad67eab36688fa2c1936b06354). New RNA-seq data generated as part of this study is deposited in NIH sequence read archive (SRA) under the project accession: PRJNA785746.

The following dataset was generated:

| Author(s) | Year | Dataset title | Dataset URL | Database and Identifier |
|---|---|---|---|---|
| Marcus S, Andrea H | 2021 | MERKO RNA-seq | https://www.ncbi.nlm.nih.gov/bioproject/PRJNA785746 | NCBI BioProject, PRJNA785746 |

The following previously published datasets were used:

| Author(s) | Year | Dataset title | Dataset URL | Database and Identifier |
|---|---|---|---|---|
| GTEx Consortium | 2015 | Genotype-Tissue Expression Project (GTEx) | https://www.ncbi.nlm.nih.gov/projects/gap/cgi-bin/study.cgi?study_id=phs000424.v8.p2 | NCBI dbGaP, phs000424.v8.p2 |
| Rubenstein AB, Smith GR, Raue U, Ruf-Zamojski F, Nair V, Zhou L, Zaslavksy E, Trappe S, Sealfon S | 2020 | Single-cell transcriptional profiles in human skeletal muscle | https://www.ncbi.nlm.nih.gov/geo/query/acc.cgi?acc=GSE130646 | NCBI Gene Expression Omnibus, GSE130646 |

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
