## [Editor Report]

This elegantly performed systems-genetics paper on the predicted human skeletal muscle secretome highlights the importance of sex and sex hormones in regulating myokine expression and predicted cross-tissue effects. Male and female mice lacking estrogen receptor α (Esr1) were used to understand how estrogen signalling affects myokine genes expression. The methods used and data presented in this manuscript can serve as an important resource for other researchers in the field.

---

## [Decision Letter]

**Decision letter after peer review:**

Thank you for submitting your article "Genetic variation of human myokine signaling is dominated by biologic sex and sex hormones" for consideration by *eLife*. Your article has been reviewed by 3 peer reviewers, one of whom is a member of our Board of Reviewing Editors, and the evaluation has been overseen by Carlos Isales as the Senior Editor. The following individual involved in review of your submission has agreed to reveal their identity: Frode Norheim (Reviewer #3).

Essential revisions:

A) Requests from reviewer #2:

The authors can address some issues:

1. On line 76, "Sex hormones are enriched with myokine expression independent of biological sex" In general, this section can be written more directly, and materials can be described in greater detail. For example, it would be better to provide the number of secreted protein gene expression studied.

2. The authors could modify the heading on Figure 1C to "Proportions" or "Sex-specific assignment of Myokine Expression."

3, Increase the font size of Fig 1D. In general, the font sizes of some of the plots in figures need to be increased.

4. Label the x-axis of Fig 1E

5. On line 103, "...hormone receptor correlations, gene expression was markedly higher in males compared to females..."

6. On line 202, "serval" -> several.

B) Requests from reviewer #3

This is an elegantly written manuscript. However, I have some recommendations for the authors.

1. I would recommend to include aim(s) in the abstract. The "background" part of the abstract can be shortened.

2. Some of the font sizes in the figures are too small. E.g. Figure 1D and 3G.

3. McMahon CD et.al. (AJPEM, 2003) shows the complexity of myostatin regulation in skeletal muscle. I would recommend that you measure myostatin in muscle and plasma in WT and Esr1 (MERKO) male and female mice. The manuscript will be improved if you can do some sort of validation from your mRNA predictions.

*Reviewer #1 (Recommendations for the authors):*

This reviewer acknowledges the interest and potential importance of genetic methodology in the resolution of physiological questions, but will defer to the more expert detailed opinions of the remaining more expert reviewers.

*Reviewer #2 (Recommendations for the authors):*

The authors can address some issues:

1. On line 76, "Sex hormones are enriched with myokine expression independent of biological sex" In general, this section can be written more directly, and materials can be described in greater detail. For example, it would be better to provide the number of secreted protein gene expression studied.

2. The authors could modify the heading on Figure 1C to "Proportions" or "Sex-specific assignment of Myokine Expression."

3, Increase the font size of Fig 1D. In general, the font sizes of some of the plots in figures need to be increased.

4. Label the x-axis of Fig 1E

5. On line 103, "...hormone receptor correlations, gene expression was markedly higher in males compared to females..."

6. On line 202, "serval" -> several.

*Reviewer #3 (Recommendations for the authors):*This is an elegantly written manuscript. However, I have some recommendations for the authors.

– I would recommend to include aim(s) in the abstract. The "background" part of the abstract can be shortened.

– Some of the font sizes in the figures are too small. E.g. Figure 1D and 3G.

– McMahon CD et.al. (AJPEM, 2003) shows the complexity of myostatin regulation in skeletal muscle. I would recommend that you measure myostatin in muscle and plasma in WT and Esr1 (MERKO) male and female mice. The manuscript will be improved if you can do some sort of validation from your mRNA predictions.

---

## [Author Response]

Reviewer #1 (Recommendations for the authors):This reviewer acknowledges the interest and potential importance of genetic methodology in the resolution of physiological questions, but will defer to the more expert detailed opinions of the remaining more expert reviewers.

We thank the reviewer for their enthusiasm and agree that this study will provide a nice foundation for investigation of muscle-derived cross-talk with other tissues. Our hope is that similar studies which provide detailed walk-throughs of analyses and readily-available data will enable such approaches to be easily accessible to the broad scientific community.

Reviewer #2 (Recommendations for the authors):The authors can address some minor issues:1. On line 76, "Sex hormones are enriched with myokine expression independent of biological sex"

We have used this new heading of the section which is much more intuitive now.

In general, this section can be written more directly, and materials can be described in greater detail. For example, it would be better to provide the number of secreted protein gene expression studied.

Thank you, this is a good suggestion and also noted by Rev #3. We have “cleaned” this section to be more direct and indicate the total number of secreted proteins used (3,666).

2. The authors could modify the heading on Figure 1C to "Proportions" or "Sex-specific assignment of Myokine Expression."

The heading has been changed on a revised Fig 1.

3. Increase the font size of Fig 1D. In general, the font sizes of some of the plots in figures need to be increased.

This point was also raised by Rev #3. The figures have also been cleaned and font-sizes increased/normalized. The pathways names in Fig 1D and revised Fig 1J-K are now more legible. The two other main figures have also been altered similarly.

4. Label the x-axis of Fig 1E

We apologize for the oversight. The x-axis has now been labelled as “Relative proportion (%)”

5. On line 103, "...hormone receptor correlations, gene expression was markedly higher in males compared to females..."

Thank you. The new sentence reads: “Among these, the master regulator of skeletal muscle differentiation and proliferation, myostatin (*MSTN*), where hormone receptor correlations and gene expression was markedly higher in males compared to females”

6. On line 202, "serval" -> several.

We apologize for the oversight and it has been modified to “several”

Reviewer #3 (Recommendations for the authors):This is an elegantly written manuscript. However, I have some recommendations for the authors.– I would recommend to include aim(s) in the abstract. The "background" part of the abstract can be shortened.

We have modified the text accordingly. Our initial mistake was to label the section as Abstract/Introduction. There are now separate Introduction and Abstract sections, where the Abstract is focused on study aims. Additionally, we have moved broad summary statements to the “*eLife* Digest” portion and narrowed the introduction significantly.

– Some of the font sizes in the figures are too small. E.g. Figure 1D and 3G.

This point was also noted by Rev #2. We have revised all figure fonts and organization to be more legible.

– McMahon CD et.al. (AJPEM, 2003) shows the complexity of myostatin regulation in skeletal muscle. I would recommend that you measure myostatin in muscle and plasma in WT and Esr1 (MERKO) male and female mice. The manuscript will be improved if you can do some sort of validation from your mRNA predictions.

Thank you for pointing out this highly relevant citation and broad limitation of the study. We have revised the manuscript in several ways to highlight this point:

1. Immunoblotting for myostatin was performed on male and female WT and MERKO mice (revised Figure 1H and supplement 2). While the sex-specificity of myostatin and differential expression following *ESR1* ablation in males was consistent with RNA-Seq, females also showed a reduction of *MSTN* related to *ESR1*. These data are included in the revised manuscript, where we utilize as a point to discuss the limitation with reliance on gene expression to infer function. It is interesting to note that the *ESR1* ~ *MSTN* correlations in GTEx were consistent with the immunoblot; however, gene expression in mice failed to predict this observation.

2. Generally, the language has been toned down in referencing correlated gene expression as “causal” in inferring signaling. This is apparent throughout the revised text, figures and title

3. The conclusion/limitations section has been revised to highlight this limitation. Specifically, we added: “Further, our current analyses rely on gene expression to guide functions of proteins which are typically strongly regulated by post-transcriptional processes. As shown for myostatin (Figure 1), gene expression analyses can miss key functional aspects of proteins, where follow-up studies and resources focused on protein and subsequent modification levels will heavily improve predictions.”